# Quercetin: A Promising Candidate for the Management of Metabolic Dysfunction-Associated Steatotic Liver Disease (MASLD)

**DOI:** 10.3390/molecules29225245

**Published:** 2024-11-06

**Authors:** Julia Markowska, Kamila Kasprzak-Drozd, Przemysław Niziński, Magdalena Dragan, Adrianna Kondracka, Ewa Gondek, Tomasz Oniszczuk, Anna Oniszczuk

**Affiliations:** 1Science Circle of the Department of Inorganic Chemistry, Medical University of Lublin, Dr. Witolda Chodźki 4a, 20-093 Lublin, Poland; juliamarkowska5@gmail.com (J.M.); magdalena.dragan@onet.eu (M.D.); 2Department of Inorganic Chemistry, Medical University of Lublin, Dr. Witolda Chodźki 4a, 20-093 Lublin, Poland; kamilakasprzakdrozd@umlub.pl; 3Department of Pharmacology, Medical University of Lublin, Radziwiłłowska 11, 20-080 Lublin, Poland; przemyslaw.nizinski@umlub.pl; 4Department of Obstetrics and Pathology of Pregnancy, Medical University of Lublin, 20-081 Lublin, Poland; adriannakondracka@wp.pl; 5Department of Food Engineering and Process Management, Institute of Food Science, Warsaw University of Life Sciences, Nowoursynowska 159C, 02-776 Warsaw, Poland; 6Department of Thermal Technology and Food Process Engineering, University of Life Sciences in Lublin, Głęboka 31, 20-612 Lublin, Poland; tomasz.oniszczuk@up.lublin.pl

**Keywords:** metabolic dysfunction-associated steatotic liver disease (MASLD), quercetin, polyphenols, nutraceuticals, functional food, oxidative stress, antioxidants

## Abstract

Metabolic dysfunction-associated steatotic liver disease (MASLD) represents a chronic liver disease. The development of MASLD is influenced by a multitude of diseases associated with modern lifestyles, including but not limited to diabetes mellitus, hypertension, hyperlipidaemia and obesity. These conditions are often consequences of the adoption of unhealthy habits, namely a sedentary lifestyle, a lack of physical activity, poor dietary choices and excessive alcohol consumption. The treatment of MASLD is primarily based on modifying the patient’s lifestyle and pharmacological intervention. Despite the absence of FDA-approved pharmacological agents for the treatment of MASLD, several potential therapeutic modalities have demonstrated efficacy in reversing the histopathological features of the disease. Among the botanical ingredients belonging to the flavonoid group is quercetin (QE). QE has been demonstrated to possess a number of beneficial physiological effects, including anti-inflammatory, anticancer and antifungal properties. Additionally, it functions as a natural antioxidant. Preclinical evidence indicates that QE may play a beneficial role in reducing liver damage and improving metabolic health. Early human studies also suggest that QE may be an effective treatment for MASLD due to its antioxidant, anti-inflammatory, and lipid-regulating properties. This review aims to summarize the available information on the therapeutic effects of QE in MASLD.

## 1. Introduction

Metabolic dysfunction-associated steatotic liver disease (MASLD) is a chronic liver disease that includes two types of liver conditions: non-alcoholic steatohepatitis and simple steatosis [1]. The disease is caused by the excessive accumulation of fat within liver cells. The development of MASLD is influenced by a multitude of diseases associated with modern lifestyles, including diabetes mellitus, hypertension, hyperlipidaemia and obesity. These conditions frequently result from the adoption of unhealthy habits, such as a sedentary lifestyle, a lack of physical activity, poor dietary choices and excessive alcohol consumption. [2,3]. Meta-analyses show that the disease affects nearly 30% of the world’s population, including 3–10% of children [4]. Studies show a higher prevalence among men than among women [5]. Considering the world’s continued development, the incidence of MASLD is expected to increase in the coming years [6]. The development of MASLD in the body can be a precursor to steatohepatitis, which can eventually lead to irreversible liver parenchymal damage and cirrhosis. MASLD is also a risk factor for the development of many metabolic diseases, such as type 2 diabetes, atherosclerosis, and liver cancer [7,8]. Treatment of MASLD is mainly based on modification of the patient’s lifestyle (weight reduction, change in dietary habits, increased physical activity) and pharmacological treatment including the optimization of parameters whose incorrect values may serve to exacerbate the disease (blood pressure, vascular permeability, blood glucose or triglyceride levels) [7]. In the treatment of MASLD, a significant role can be noticed for food and the active ingredients present in it, especially those of natural origin [9]. Although there are currently no drugs approved by the US Food and Drug Administration (FDA) for treatment, several potential therapies have been shown to be effective in reversing the histopathological features of the disease [7].

The term NAFLD (non-alcoholic fatty liver disease) has been used to describe the histological spectrum from steatosis to steatohepatitis. However, the term NAFLD has some limitations as it is based on exclusionary confounding terms and uses potentially stigmatizing language. In 2023, it was decided that it would be appropriate to change the acronym to MASLD, which means ‘metabolic dysfunction-associated steatotic liver disease’ [10,11].

One of the botanical ingredients that belongs to the flavonoid group is quercetin (3,3′,4′,5,7-pentahydroxyflavone). Quercetin (QE) has a polyphenolic structure with an oxygen substituent at position 4 and a double bond between the 2nd and 3rd carbon atoms [12]. It is a naturally occurring flavonoid in fruits and vegetables such as tomatoes, onions and berries, in the leaves of green plants and in red wine too. QE exhibits a number of health-promoting properties, such as anti-inflammatory, anticancer and antifungal effects. It is also a natural antioxidant [13]. The chemical structure of QE is shown in Figure 1.

Scientific studies have shown the beneficial effects of QE aglycone at a dose of 500 mg on reducing high blood pressure and inflammation in the body. After absorption, the substance is metabolized by liver cells to glucuronides, which are then excreted by the kidneys in the urine [14].

Preclinical evidence supports a beneficial role for QE in reducing liver damage and improving metabolic health. Early human studies are also encouraging. Eating foods rich in QE, such as apples, onions and berries, may be a safe and natural approach to supporting liver health. QE shows promise in the treatment of MASLD due to its antioxidant, anti-inflammatory and lipid-regulating properties. This review summarizes the available information on the therapeutic effects of QE in MASLD [2].

## 2. Pathogenesis of MASLD

MASLD has often been perceived as a hepatic manifestation of metabolic syndrome (MetS), also known as “X syndrome” [15]. A schematic representation of connection between components of MetS and MASLD is shown in Figure 2.

Based on initial studies, the ‘two-hit’ hypothesis was formulated and widely accepted [16]. As the primary causes of MASLD pathogenesis, sedentary lifestyle leading to abdominal obesity as well as an improper, fat-rich diet and the developing of insulin resistance (IR) were perceived, later so-called the ‘first-hit’. As a consequence of the ‘first-hit’, a number of inflammatory processes leading to fibrogenesis have to be activated, later called as the ‘second-hit’ [17,18]. However, as the evidence of a very complex, multi-dimensional relationship between traditional risk factors of MASLD and the progression of disease is grew, a novel theory of MASLD pathogenesis must have been proposed. So in light of the multifaceted cause-and-effect relationship of MASLD development, the ‘multiple-hit’ hypothesis were formulated [19]. According to the novel but widely accepted ‘multiple-hit’ doctrine, the most probable causes of MASLD related to quercetin activity are briefly delineated in the next subsections.

### 2.1. Lipid Metabolism and Insulin Signalling

Excess energy intake activates certain metabolic pathways leading to lipid formation and deposition, mainly in subcutaneous white adipose tissue (WAT). In the course of liver steatosis and its progression to MASLD and MASH (metabolic dysfunction-associated steatohepatitis) these mechanisms are impaired, leading to ectopic fat accumulation, mainly in hepatocytes [20]. Lipids are stored in liver tissue mainly in the form of triglycerides, which are synthesized from free fatty acids (FFAs). The most likely reasons for the excessive supply of FFAs in the bloodstream and consequent overabundance in the liver are a high-fat, high-sugar diet, de novo lipogenesis (DNL) in hepatocytes and high levels of lipids stored in WAT, particularly in obese individuals [21]. In addition to dietary fat intake and the transport of FFAs from WAT, de novo lipogenesis is also considered to be a very important source of FFAs in hepatocytes and is closely related to insulin resistance and carbohydrate overloading. IR seems to play a crucial role in altering lipid metabolism as insulin is characterized with anti-lipolytic effects. It also contributes to the maintenance of TG accumulation in WAT and the storage of FFAs, resulting in decreased levels of TGs and FFAs in the liver [22]. IR and the associated downregulation of insulin receptor substrate 2 (IRS-2) leads the to overexpression of sterol regulatory element-binding protein 1c (SREBP-1c) and carbohydrate response element-binding protein (ChREBP), two transcription factors involved in the expression of genes encoding the major enzymes mediating DNL: fatty acid synthase (FAS) and acetyl-CoA carboxylase (ACC) [23]. The upregulation of DNL during IR leads to the synthesis of excessive amounts of FFAs and their further conversion to TGs and accumulation in the liver [24]. In addition to the excessive influx and synthesis of FFAs in the liver, alterations in the transport of lipids from hepatocytes to peripheral tissues were also reported, indicating the complexity of maintaining a balance between hepatic lipid input and output [25].

Peroxisome proliferator-activated receptor gamma (PPARγ) is a nuclear receptor transcription factor that plays a crucial role in lipid metabolism, adipogenesis, and insulin sensitivity. While its primary function is associated with adipose tissue, emerging evidence links PPARγ to the development of MASLD. The ability of this receptor to regulate lipid uptake, storage, and metabolism has implications in hepatic lipid accumulation and the pathogenesis of steatosis. Under pathological conditions like obesity, insulin resistance, and metabolic syndrome, PPARγ expression is significantly upregulated in hepatocytes. Studies showed that increased hepatic PPARγ expression correlates with enhanced lipid storage, leading to steatosis. Importantly, the upregulation of PPARγ in the liver is driven by various factors, including inflammatory cytokines, dietary factors (high-fat diets), and metabolic stress [26].

PPARγ activation enhances the expression of genes involved in fatty acid transport and lipid storage, such as CD36 (a fatty acid transporter) and the adipocyte fatty acid-binding protein (FABP4). In hepatocytes, PPARγ promotes the uptake of circulating free fatty acids and their subsequent esterification into triglycerides for storage. This process leads to an increase in intracellular lipid droplets. It also regulates genes involved in lipogenesis, including SREBP-1c, which drives de novo fatty acid synthesis. PPARγ is well-known for its role in improving insulin sensitivity, particularly in adipose tissue. However, in the liver, the relationship between PPARγ and insulin sensitivity is more complex. While agonists of this receptor, improve systemic insulin sensitivity, chronic PPARγ activation in hepatocytes can paradoxically promote insulin resistance. Although PPARγ primarily promotes lipid storage, it also has anti-inflammatory properties. The activation of this receptor reduces the expression of pro-inflammatory cytokines, such as TNF-α (tumor necrosis factor-alpha) and IL-6 (interleukin 6), in both adipose tissue and the liver.

PPARγ has emerged as a potential therapeutic target for treating fatty liver disease due to its involvement in lipid metabolism and insulin sensitivity. Targeting PPARγ with selective modulators holds promise for treating MASLD, though balancing efficacy and safety remains a critical challenge for future drug development [27].

### 2.2. Oxidative Stress, Inflammation and Lipotoxicity

Under normal conditions, FFAs derived from both DNL and the influx from peripheral tissues to hepatocytes are the source for the synthesis of TGs, which in turn are stored in liver cells and can be used as an energy reservoir in times of energy deficit in the body. Energy in the form of ATP is produced from both TGs and FFAs by the β-oxidation process [28]. In a healthy liver, β-oxidation is maintained by numerous regulatory factors so that the free electrons generated from lipid oxidation are neutralized, ultimately resulting in a neutral compound—water (H_2_O). When liver cells are overloaded with lipids, the electron transport chain may be disturbed and unable to neutralize a significant amount of free electrons, resulting in the production of reactive oxygen species (ROS), mainly in the form of hydrogen peroxide (H_2_O_2_) or superoxide (O_2_**^.^**) [29,30]. ROS are essential for maintaining the optimal physiological function and contributing to the development of MASLD [31]. ROS facilitate the immune response and the transmission of signals within cells. They are produced during physiological processes in macrophages, endothelial cells or pneumocytes. However, when produced in excess, they lead to oxidative stress and damage to lipids, proteins and DNA. As a consequence of peroxidation, the structure of biomolecules undergoes alteration, resulting in the loss or modification of their intrinsic functions [32]. Biological membranes contain significant quantities of PUFAs, namely polyunsaturated fatty acids, rendering them susceptible to attack by free radicals. As a consequence of their interaction with other molecules, they undergo a transformation into free radicals, which possess the capacity to instigate a cascade of lipid peroxidation reactions. The products of these reactions possess high biological activity, with the potential to destroy DNA, proteins, inhibit enzyme activity and initiate cell death by activating signalling pathways [33].

The cause and effect relationship between abnormal lipid levels in the liver and their adverse consequences was termed ‘lipotoxicity’ [34]. Oxidative stress is responsible for the increased expression of many pro-inflammatory molecules, including TNF-α, some interleukins (IL-6 and IL-1β) or transforming growth factor beta (TGF-β). In addition, the lipoxygenase (LOX) and cyclooxygenase (COX) pathways are also upregulated under oxidative stress, suggesting that chronic inflammation caused by lipotoxicity is a trigger for histopathological changes in the course of MASLD and its progression to MASH [29,35].

### 2.3. Autophagy

Autophagy is a complex set of processes aimed at maintaining cellular homeostasis by transporting, degrading and recycling damaged or dysfunctional small molecules, proteins or even entire organelles. The exact mechanism of autophagosome formation is still not fully understood, however the theories of derivation from endoplasmic reticulum or mitochondria have been proposed [36]. In general, the regulation of autophagy is maintained by several molecular pathways, with the crucial role of autophagy-related proteins (Atg) [37]. There is some evidence that autophagy may be impaired in individuals with MASLD [38,39]. Under physiological conditions, autophagy is responsible for both the degradation and formation of lipid droplets (LDs). Lipid droplets are organelles in hepatocytes where TGs are stored and serve as a reservoir for further use in various metabolic pathways [40]. Autophagy mediates the hydrolysis of intracellular TGs to FFAs and assists in their efflux from LDs and hepatocytes to peripheral tissues, alleviating liver steatosis [41,42,43].

One of the subtypes of autophagy, targeting mitochondria is the so-called process ‘mitophagy’. Under physiological conditions mitophagy facilitates proper mitochondria function due to the removal of damaged, dysfunctional or even excessive numbers of these organelles [44]. Since mitochondria serve as energy converting structures, they are exposed to ROS, as a by-product of ATP synthesis [45]. Moreover, when cells are overloaded with lipids, an excessive amount of ROS is generated, which in turn leads to oxidative stress and inflammation [30]. Mitophagy plays crucial role in the identification and exclusion of damaged mitochondria, but certain signalling molecules are essential in order to mark damaged organelles and allow them to be disposed [46]. The most studied regulatory cascade of mitophagy is the PINK1/PARKIN signalling pathway. PINK1 is Ser/Thr kinase, which in normal conditions is constantly cleaved and degraded by PARL protease. In damaged mitochondria PINK1 is accumulated in the outer membrane of mitochondria due to its depolarization, which allows for the phosphorylation of ubiquitin and E3 ubiquitin ligase, namely PARKIN, which in turn serve as signalling molecules activating the mitophagy process [45,46]. Recent studies on both in vitro and in vivo show that reduced mitophagy may be an early sign of MASLD, yet the possible molecular mechanisms involve the upregulation of ZNF143/lncRNA NEAT1/ROCK2 axis which leads to the reduced expression of PINK1 and PARKIN, thus impairing mitophagy [47,48].

On the other hand, impaired autophagy could lead to liver injury in a different manner. For instance, in Atg7-knockout mice it was observed that autophagy could protect liver cells from lipopolysaccharide (LPS)-induced damage [49]. Furthermore, hepatocytes were more susceptible to the inflammatory processes in Atg5 or Atg7-knockout mice [50]. Therefore, it is possible that autophagy is not only the primary process by which stored triglycerides are converted to free fatty acids in the liver, but also has a protective effect on the hepatocytes via different pathways [51,52].

### 2.4. Gut Microbiota

The gut microbiota plays a multifaceted role in the pathogenesis of NAFLD by influencing liver metabolism, inflammation and insulin sensitivity. The term ‘microbiota’ refers to the entire collection of bacteria, archaea and eukarya found in the gastrointestinal (GI) tract, including approximately 10–100 trillion microorganisms [53]. The gut microbiota is essential for the maintenance of many processes in the host organism, such as facilitating the metabolism of selected nutrients and drugs, assisting in immune responses or maintaining the integrity of the gut lumen [54]. As the main source of blood for the liver, it comes from the portal vein, but many beneficial substances produced in hepatocytes are absorbed by the gut. It has been shown that there is a close relationship and communication between the gut and the liver [55].

The development and diversifying of the gut microbiota just from birth and early life is perceived as critical in the formation of proper microbiome composition [56]. Under physiological conditions, the gut microbiota has a certain composition of microbes, but with an excessive intake of some dietary products (e.g., fructose), the ratio of Firmicutes to Bacterioidetes could be altered. An increased amount of Firmicutes in the gut lumen has been associated with an increased risk of T2DM (type 2 diabetes) or obesity, which in turn affects the development of MASLD. Furthermore, Bacterioidetes are considered to be a neutral or rather positive phylum that can produce many beneficial substances [57]. Furthermore, it has also been confirmed, that the improper development of gut microbiota in childhood can lead to obesity in later years [56]. Nevertheless, microbial-derived molecules can stimulate numerous receptors and/or affect various metabolic pathways. One of the most studied bacterial-derived substances affecting liver metabolism are short-chain fatty acids (SCFAs), in particular propionate, butyrate and acetate. The typical ratio of these compounds in the gut is 1:1:3, respectively [58]. SCFAs are considered to be a source of energy for hepatocytes [59], important molecules that help maintain the proper function of the intestinal barrier [60], anti-inflammatory and immunomodulating agents [61] or regulators of bile acid (BA) secretion [62]. It has been reported that alterations in the ratio of SCFAs in the gut can have serious consequences. For example, increased intestinal permeability and increased bacterial endotoxin (e.g., lipopolysaccharide, LPS) flux to the liver [63] lead to the induced expression of pro-inflammatory cytokines (TNF-α, IL-6, IL-1β) [64]. LPS after its absorption and translocation to the hepatocytes is recognized by toll-like receptor 4 in the cellular membrane of hepatocytes, which leads to upregulate the nuclear factor kappa-light-chain-enhancer of activated B cells (NF-κB) and the overexpression of pro-inflammatory molecules [65]. In addition, changes in gut microbiota composition can disrupt the metabolism of BAs through the farnesoid X receptor (FXR) and Takeda G protein-coupled receptor 5 (TGR5) signalling pathways, leading to abnormal carbohydrate and lipid absorption and subsequent metabolic responses [66]. The improper composition of gut microbiota and related alterations in SCFAs levels can also lead to slowed gut motility, mainly by the excessive secretion of peptide YY (PYY) [67]. Another explanation of reduced GI motility is the damaging and loss of enteric neurons, caused primarily by LPS-induced neurotoxicity [68] and lipotoxicity [69].

The role of the gut microbiota in relation to the development of MALSD, and ultimately the use of probiotics to alleviate MALSD, is becoming an area of great interest, and understanding the exact role of specific microorganisms in the pathogenesis of MALSD is currently one of the most studied issues. An updated, comprehensive review of the role of microbiota in MASLD has recently been published elsewhere [70].

## 3. Mechanism of Action of Quercetin in MASLD—In Vitro and In Vivo Studies

Quercetin has been demonstrated to possess a range of biological activities, including antioxidant, anti-inflammatory, antibacterial [71], antiviral, anticancer [72] antiproliferative and gene expression modifying capabilities in vitro. These mechanisms are fundamental to the beneficial impact of this compound on disease processes [73]. The principal mechanisms of action of quercetin in the treatment of metabolic syndrome and associated disorders are outlined below.Quercetin is characterized by strong antioxidant properties [73,74] through its ability to remove free radicals and ROS. QE can stop the spread of lipid peroxidation, as it has the ability to increase glutathione levels [2,75]. Furthermore, it was demonstrated that QE resulted in an enhancement in the activity of superoxide dismutase and catalase. [31]. QE affects the total antioxidant capacity and consequently increases endogenous antioxidant activity through significant benefits. QE reduces oxidative stress induced by sodium fluoride and, consequently, is beneficial for liver function in the rats studied [76]. It is postulated that quercetin may also act by stimulating cellular defence against oxidative stress, which is presumed to be due to its direct antioxidant activity. The pathways that are responsible for this activity are the induction of Nrf2-ARE and the induction of the antioxidant/anti-inflammatory enzyme paraoxonase 2 (PON2) [77]. A schematic representation of the beneficial effects of QE in MASLD with selected molecular mechanisms are shown in Figure 3.

### 3.1. Reduction in Oxidative Stress

A number of factors are involved in the progression of MASLD in a parallel and cooperative manner [78]. The protective role of QE in MASLD has been investigated in numerous experimental models. Several pathways have been implicated, highlighting the complexity of QE action in cellular function and physiology.

The ability of QE to scavenge free radicals has been established for some time. Experimental studies demonstrated that QE treatment elevates the levels of antioxidant enzymes, including catalase (CAT), superoxide dismutase (SOD), and glutathione (GSH), in the livers of leptin receptor mutated mice and reduces ROS in cells subjected to various damaging agents [79]. In studies on cell lines (e.g., L02 cells, BRL-3A hepatocytes, HepG2 cells), quercetin and quercetin-3-O-β-D-glucuronide were observed to antagonize oxidative stress and inflammation induced by different compounds [80]. In an in vitro study, Vidyashankar et al. [31] subjected HepG2 cells to steatosis through incubation with oleic acid. The researchers observed a notable elevation in the levels of the pro-inflammatory cytokines TNF-α and IL-8 following oleic acid administration. Furthermore, there was an observable inhibition of glucose uptake and cell proliferation. QE has been demonstrated to enhance insulin sensitivity, a pivotal aspect of MASLD. Additionally, the accumulation of fat was diminished, and cell proliferation was enhanced. Furthermore, the inhibition of IL-8 and TNF-alpha levels with augmented cellular glutathione was achieved. The authors concluded that QE effectively reversed the symptoms of MASLD by increasing the secretion of cellular antioxidants and reducing triacylglycerol accumulation, insulin resistance, and the level of inflammatory cytokines.

In a study conducted by Zhang et al. [79] it was demonstrated that QE has the capacity to alleviate oxidative stress by reducing the total amount of ROS, hydrogen peroxide, and by increasing the activity of SOD, while simultaneously inhibiting the NLRP3 inflammasome in rats. Furthermore, the inhibition of the ROS and pathway and thioredoxin interacting protein (TXNIP) has been shown to demonstrate the protective effect of QE. However, there is a paucity of evidence elucidating the mechanisms through which QE prevents oxidative stress in MASLD, and the detailed molecular mechanisms remain unclear.

Surapaneni et al. [81] demonstrated that QE provides protection to the liver against experimentally induced non-alcoholic steatohepatitis in rats. An experimental model of steatohepatitis in rats was developed by providing the animals with an HFD for eight weeks. This model was subsequently employed in a comparative study investigating the role of QE on a variety of parameters associated with steatohepatitis. QE can decrease cytochrome P450 2E1 (CYP2E1) levels, thereby reducing oxidative stress mediated by CYP2E1.

### 3.2. Regulatory Effects on Inflammatory Pathways

The development of the inflammatory process in the MASLD process is associated with oxidative stress, hepatic stellate cell activation, apoptosis, and other pathways [12]. The beneficial effects of QE on MASLD are mediated by its ability to modulate a number of inflammatory pathways. Quercetin maintains inflammatory balance in HFD-fed mice. One study showed that this compound inhibits the nuclear import of NF-κB (p65) [82]. Hepatic steatosis triggers the activation of NF-κB and this action is the cause of the body’s increased production of pro-inflammatory cytokines such as TNF-α, IL-6 and IL-1β [74]. Among the pro-inflammatory biomarkers, induced nitric oxide synthase (iNOS) and interferon gamma (IFN-γ) are solely dependent on NF-κB, with the induction of both genes following activation of NF-κB. Induced by a high-fat diet, ROS are involved in the release of both Th1 cytokines, TNF-α [82]. This factor is a pleiotropic pro-inflammatory cytokine and at the same time one of 22 proteins belonging to the TNF superfamily, regulating cell differentiation and growth [83] and (IL-6. In this study, QE also decreased the expression of iNOS, IFNγ and CRP (C-reactive protein) genes [82] (CRP is a marker of, among other things, inflammation [84]) and the release of serum inflammatory cytokines TNF-α and IL-6 [82]. The administration of QE to non-alcoholic steatohepatitis rats blocked NLRP3 inflammasome activation, inhibited inflammation and impaired lipid metabolism. Antioxidant interventions can inhibit the ROS-TXNIP pathway [74]. QE exhibited hepatoprotective activity by regulating inflammation-related metabolites (12(S) -HPETE and arachidonic acid) [85].

The mechanisms underlying the beneficial anti-inflammatory effects of QR in liver injury result from the activation of novel pathways such as the decreased expression of the Toll 4 receptor (TLR4), neutrophil elastase (NE) and the P2Y2 purinergic receptor. This downregulation leads to reduced apoptosis and liver fibrosis without altering hepatocyte proliferation [86]. Due to its anti-inflammatory and antioxidant properties, quercetin reduces fat accumulation, which affects the occurrence and development of MASLD [74].

Quercetin affects fibrotic processes in the liver. An important mechanism of this effect is anti-inflammatory activity. A study was carried out to determine the effect of QE on macrophage activation and polarization. This compound led to a significant alleviation of liver inflammation and fibrosis, and the inhibition of hepatic stellate cell activation. QE inhibited macrophage activation and M1 polarization and reduced the mRNA expression of M1 macrophage markers such as TNF-α, IL-1β, IL-6 and nitric oxide synthase 2. Quercetin inhibition of M1 macrophages was associated with reduced levels of Notch1 expression on macrophages [16].

In animal models, QE has been shown to reduce inflammation and improve liver health in MASLD. HFD-induced MASLD rat models treated with QE demonstrated significant reductions in liver enzymes such as alanine aminotransferase (ALT) and aspartate aminotransferase (AST). QE also alleviated lipid accumulation and steatosis by modulating lipid metabolism and oxidative stress pathways, primarily through the upregulation of fatty acid oxidation via the PPARα pathway. It further reduced hepatic inflammation by inhibiting macrophage infiltration, but the effects were more pronounced in the early stages of MASLD, with a noticeable drop in inflammation over 30 days compared to 50 days of treatment [85].

### 3.3. Regulation of Lipid Metabolism and Mitochondrial Dysfunction

Mitochondrial dysfunction is closely linked to the progression of MASLD. Quercetin may be involved in aspects related to the development of MASLD through the interplay or parallelism of numerous mechanisms. Areas in which it is involved include metabolism and formation of lipids and proteins, bile secretion, cellular activities, involvement in signalling pathways, as well as drug and xenobiotic metabolism (cytochrome P450) [12]. Disorders of lipid homeostasis may result from alterations in the pathways involved in lipid metabolism. The accumulation of lipids in the liver may be the cause of the increased uptake of FFAs by hepatocytes. Elevated blood levels of FFAs contribute to fatty liver and promote excessive TG synthesis [10].

A gene expression profiling study showed that QE increased hepatic lipid metabolism (mainly due to ω-oxidation) and decreased the corresponding levels of circulating lipids. At the gene level, this action resulted in the increased expression of cytochrome P450 (Cyp) 4a10, Cyp4a14, Cyp4a31 and acyl-CoA 3 thioesterase (Acot3) and regulators: cytochrome P450 oxidoreductase (Por, which limits the rate of cytochrome P450) and the constitutive androstane receptor transcription factor [87]. Another mechanism of action may be the modulation of gene expression associated with lipid metabolism by inactivating the PI3K/Akt pathway, reducing the expression of two de novo lipogenesis genes, including SREBP-1c and LXRα, and the gene associated with fatty acid uptake and transport, fatty acid translocase CD36 (fat/CD36). In addition, QE increases the expression of fatty acid transporting protein 5 (FATP5), fatty acid binding protein 1 (FABP1), the forkhead box A1 protein (FOXA1) and the small heterodimer partner (SHP) [88]. It is worth noting that the expression of hepatic genes associated with steatosis, such as peroxisome proliferator-activated receptor gamma and sterol-1c regulatory element-binding protein, is also normalized by QE [89]. The anti-obesity effect of QE may be associated with the regulation of lipogenesis at the level of transcription. QE supplementation altered the expression profiles of several lipid metabolism-related genes, including Fnta, Pon1, Pparg, Aldh1b1, Apoa4, Abcg5, Gpam, Acaca, Cd36, Fdft1, and Fasn, compared to the control group [90]. Quercetin ameliorates hepatic TG deposition by promoting VLDL formation and lipophagy. QE can reduce the accumulation of lipids in the liver by promoting hepatic VLDL (very-low-density lipoprotein) assembly and lipophagy via the IRE1α/XBP1s pathway (transmembrane kinase/endoribonuclease 1α/X-box binding protein 1) requiring inositol [91]. Another mechanism by which QE counteracts the deposition of low-density lipoproteins is by inhibiting the expression of the mRNA and proteins CD36 and MSR1, which were upregulated by the high-fat diet [92].

The beneficial effects of QE on impaired glucose and lipid metabolism are probably related to the upregulated activity and protein levels of SIRT1 (sirtuin 1) and its influence on the Akt signalling pathway. Akt is a serine/threonine protein kinase that is activated by phosphorylation and deacetylation [93]. It is considered that metabolites play a significant role in the occurrence of diseases. QE showed hepatoprotective effects by regulating adrenic acid, docosahexaenoic acid, palmitic acid, oleic acid and eicosapentaenoic acid [85].

In primary lipid-loaded hepatocytes, QE has been demonstrated to upregulate pivotal markers of mitochondrial biogenesis, including peroxisome proliferator-activated receptor gamma coactivator 1-alpha (PGC-1α), PPARα and carnitine palmitoyltransferase 1-alpha (CPT-1α) [94]. These markers are of great importance for the regulation of fatty acid oxidation and energy metabolism in the liver. Furthermore, the effect of QE on mitochondrial biogenesis was found to be abolished upon the introduction of the HO-1 (heme oxygenase-1) inhibitor or Nrf2 deficiency, which provides additional evidence that the modulation of mitochondrial function by QE is mediated through the Nrf2/HO-1 pathway. The capacity of quercetin to regulate mitochondrial homeostasis has been associated with its activation of the Nrf2/HO-1 pathway. In studies involving hepatocytes lacking Nrf2, QE’s ability to restore mitochondrial function was significantly impaired, indicating that this pathway is a vital component of QE’s protective effects [94]. Activation of the Nrf2/HO-1 pathway has been demonstrated to reduce oxidative stress and increase mitochondrial biogenesis, as evidenced by the up-regulation of key markers of mitochondrial biogenesis. This dual action, which reduces oxidative damage while promoting the production of new, healthy mitochondria, highlights the potential therapeutic value of QE in combating mitochondrial dysfunction in MASLD. sincemitochondrial dysfunction is a principal factor in disease progression. By modulating mitochondrial biogenesis and reducing oxidative stress, quercetin represents a promising avenue for the prevention and potential reversal of disease progression [95]. However, further research is required to confirm the specific protective mechanisms of QE in MASLD and to determine optimal therapeutic doses and delivery methods in liver disease.

### 3.4. Regulation of Autophagy

Autophagy represents a fundamental cellular process that plays a critical role in maintaining liver homeostasis. This action is achieved by the removal of defective organelles, misfolded proteins and dysfunctional cellular components. In liver diseases such as MASLD, where excessive lipid accumulation and metabolic stress are prominent, autophagy becomes especially important in maintaining cellular health. A number of studies demonstrated that QE is capable of modulating autophagy in both in vitro and in vivo models of liver disease. One notable mechanism by which QE exerts its hepatoprotective effects is through the regulation of macroautophagy. In MASLD, impaired autophagic function represents a significant contributing factor to the accumulation of lipid droplets in hepatocytes. A research study conducted by Liu et al. [92] demonstrated that QE is capable of improving macroautophagy dysfunction, thereby facilitating the restoration of normal autophagic flux and the reduction in hepatic lipid burden. The results demonstrate that the administration of QE to mice (24 weeks) resulted in a significant alleviation of liver damage induced by an HFD (high-fat diet), accompanied by a reduction in hepatic cholesterol and ox-LDL levels. The administration of QE resulted in a notable inhibition of both mRNA and protein expression of CD36 (reduced by 53% and 71%, respectively) and MSR1 (reduced by 25% and 45%, respectively), which were upregulated by the HFD. Furthermore, the expression of LC3II was upregulated 2.4 times, whereas that of glycoprotein p62 and the mammalian target of rapamycin (mTOR) were downregulated by 57% and 63%, respectively, by QE treatment. Additionally, beyond its impact on macroautophagy, QE also exerts influence over mitophagy, a distinctive form of autophagy that targets impaired or dysfunctional mitochondria for degradation. Mitophagy ensures the clearance of damaged mitochondria, preventing the release of ROS that can exacerbate liver injury.

In a study published in 2018, Liu et al. [96] demonstrated that QE enhances mitophagy through a frataxin-mediated pathway that involves PTEN-induced putative kinase 1 (PINK1) and Parkin, which are key regulators of mitochondrial quality control. Frataxin, a mitochondrial protein, is indispensable for the maintenance of mitochondrial iron homeostasis and the prevention of oxidative damage. By increasing frataxin levels, QE facilitates the PINK1/Parkin-dependent mitophagic process, thereby enabling the removal of damaged mitochondria and protecting the liver from oxidative stress and mitochondrial dysfunction. The study involved adult male C57BL/J mice fed an HFD (60% of energy from fat) with or without QE for a period of 10 weeks. The administration of QE was observed to alleviate the histopathological changes, the disorders of lipid metabolism and mitochondrial damage induced by the HFD. QE was observed to inhibit the suppression of mitophagy induced by the HFD. It was evidenced by the increased expression of LC3II and PINK1, and the decreased expression of protein p62. Moreover, QE was also observed to facilitate Parkin translocation to mitochondria, a process confirmed by immunofluorescence analysis. Specifically, frataxin levels were decreased in the livers of mice fed an HFD or HepG2 cells incubated with oleate/palmitate. However, QE treatment restored frataxin levels. The regulation of frataxin by QE may depend on protein p53. This finding highlights the importance of QE in enhancing the liver’s capacity to mitigate mitochondrial-related damage, which is a critical aspect of managing MASLD. Another crucial process in maintaining hepatic lipid homeostasis is lipophagy, a selective form of autophagy that degrades lipid droplets. The research conducted by Zhu et al. [91] demonstrated that QE activates lipophagy through the inositol-requiring enzyme 1 alpha/X-box binding protein 1 (IRE1α/XBP1s) signalling pathway. IRE1α is a pivotal regulator of the unfolded protein response, which is triggered in response to endoplasmic reticulum (ER) stress. In conditions of metabolic stress, such as in MASLD, QE induces the activation of IRE1α, which in turn leads to the splicing of XBP1s, a transcription factor that enhances autophagy and lipophagy. Male Sprague-Dawley rats were fed an HFD, and HepG2 cells were stimulated with free fatty acids. The objective was to explore the effect of signalling pathways involved in endoplasmic reticulum stress on VLDL assembly and lipophagy. To this end, the samples were treated with QE and various pharmacological reagents. The administration of QE was observed to result in a 39% reduction in hepatic triglyceride content, accompanied by a 1.5-fold increase in VLDL level. Additionally, spliced XBP1 expression was observed to be upregulated in comparison to the HFD group. The administration of thapsigargin or STF-083010 (an IRE1α endonuclease inhibitor) resulted in a dose-dependent reduction in VLDL content, which partially counteracted the protective effects of QE, 4-PBA or APY-29 (an IRE1α endonuclease activator). An increased expression of the microsomal triacylglycerol transfer protein complex was observed in the presence of QE, while a decreased expression was observed in the presence of STF-083010. Moreover, quercetin facilitated the colocalization of lysosomes with lipid droplets, which was accompanied by a reduction in p62 accumulation. These findings demonstrate that the primary targets of QE in the treatment of MASLD via the IRE1a/XBP1s pathway are hepatic VLDL assembly and lipophagy.

Although a majority of the evidence suggests that QE exerts beneficial effects on autophagy in the liver, some studies reported findings that are contrary to this hypothesis. In a study conducted by Wu et al. [97] it was demonstrated that QE, in conjunction with its metabolite isorhamnetin, exerts an inhibitory effect on autophagy in a mouse model of CCl_4_-induced liver fibrosis. This reduction in autophagic activity was associated with the prevention of fibrosis progression. The authors proposed that the inhibitory effects of QE on autophagy in this context may be attributed to the suppression of the TGF-β1 and Smads signalling pathway, a pivotal regulator of fibrogenesis. TGF-β1 is understood to facilitate the development of fibrosis by stimulating the activation of hepatic stellate cells (HSCs), which are responsible for the production of extracellular matrix proteins that contribute to the formation of fibrotic tissue. By inhibiting the TGF-β1 signalling pathway, QE may reduce the activation of HSCs and, consequently, the development of fibrosis. This apparent duality in QE’s effects on autophagy—namely, the promotion of autophagy in the context of MASLD and the inhibition of it in fibrosis—serves to illustrate the complexity of autophagic regulation in the liver. The precise mechanisms by which QE modulates autophagy in different pathological states remain to be fully elucidated, and further research is required to clarify the conditions under which QE’s effects on autophagy are either beneficial or inhibitory.

### 3.5. Modulation of Gut Microbiota Composition

The gut-liver axis is the bidirectional relationship between the gut microbiota and liver function. The link between gut microbiota (GM) composition and BAs in MASLD is a critical area of research. The gut microbiota plays a crucial role in modifying primary BAs (cholic acid and chenodeoxycholic acid) into secondary bile acids (deoxycholic acid and lithocholic acid). This conversion happens mainly in the colon through microbial enzymes such as bile salt hydrolases and 7α-dehydroxylase. MASLD is associated with an imbalance in the gut microbiota [66]. Dysbiosis can alter bile acid metabolism, leading to an altered bile acid pool, which in turn affects liver health. MASLD patients often exhibit reduced microbial diversity, particularly in bile-metabolizing bacteria like Bacteroides and Clostridium species. These changes result in an altered ratio of primary to secondary BAs. Moreover, dysbiosis may lead to higher concentrations of toxic secondary bile acids which can have pro-inflammatory and hepatotoxic effects, promoting liver damage and contributing to steatosis and fibrosis in MASLD [60]. Moreover, bile acids act as signalling molecules that regulate metabolism through the activation of specific receptors such as FXR and TGR5. In MASLD, gut microbiota-driven changes in bile acid composition can lead to the following: intestinal barrier dysfunction and bile acid pool imbalance. Understanding the relationship between gut microbiota, bile acids, and MASLD opens up potential therapeutic avenues [66].

Numerous types of gastrointestinal bacteria have the ability to convert complex carbohydrates into secondary metabolites such as SCFAs. Propionate, butyrate, and acetate are seen as the most important in various cellular processes. A sufficient and adequate amount of SCFAs in the intestinal/hepatic axis has a beneficial effect in the course of MASLD. The appropriate Firmicutes/Bacterioides ratio is also crucial [98]. A schematic diagram of the mechanism of action of dietary polyphenols during MASLD is shown in Figure 4 [99,100].

QE has the ability to modulate GM thus contributing to counteracting obesity and reducing the degree of symptoms of non-alcoholic fatty liver disease. intestinal microbiota transplantation from a non-responsive HFD donor and a donor fed an HFD diet with the highest response to quercetin results in a protective phenotype against HFD-induced MASLD, in a mechanism that involves blocking the alteration of the enterohepatic axis in these recipients [101]. The level of Akkermansia genus in the protection from obesity-associated MASLD development turned out to be important [102]. Akkermansia muciniphila has been considered a potential probiotic due to its protective effect in obesity development [103]. Dietary intervention with Akkermansia muciniphila and quercetin supplementation reshapes gut microbiota composition in an in vivo model of early obesity related non-alcoholic fatty liver disease [104]. QE restored intestinal microbiota imbalance with HFD-induced MASLD and associated the endotoxemia-dependent induction of the TLR-4 pathway, with a subsequent inhibition of the inflammatory response and blocking of the deregulation of lipid metabolism gene expression [101]. QE and its glycoside can effectively regulate the composition of the gut microbiome, such as the genera Akkermansia, Bifidobacterium, and Lactobacillus [105].

The objective of the study by Porras et al. [101] was to ascertain the impact of experimental QE treatments on the gut microbial balance and the associated activation of the gut-liver axis in a dietary animal model of obesity-related MASLD. C57BL/6J mice were placed on a HFD, either with or without QE, for a period of 16 weeks. The administration of a HFD resulted in the induction of obesity, metabolic syndrome and hepatic steatosis. Increased intrahepatic lipid accumulation was associated with altered expression of genes related to lipid metabolism, resulting from the deregulation of their major modulators. The administration of QE resulted in a reduction in insulin resistance and MASLD activity, accompanied by a decrease in intrahepatic lipid accumulation. This was achieved through the modulation of gene expression associated with lipid metabolism CYP2E1-dependent lipoperoxidation, and associated lipotoxicity. Metagenomic analysis revealed that the HFD induced dysbiosis, as evidenced by an increase in the Firmicutes/Bacteroidetes ratio and Gram-negative bacteria, along with a marked elevation in the prevalence of the genus Helicobacter. Dysbiosis was accompanied by endotoxemia, intestinal barrier dysfunction and an altered gut-hepatic axis, which in turn resulted in the overexpression of inflammatory genes. The dysbiosis-mediated activation of TLR-4 NF-κB signalling pathway has been associated with the initiation of the inflammasome response and the induction of the reticulum stress pathway. Administration of QE reversed the imbalance in the gut microbiota and the associated TLR-4 pathway induction resulting from endotoxaemia, and subsequently inhibited the inflammasome response and reticulum stress pathway activation. As a result, the deregulation of gene expression involved in lipid metabolism was blocked.

In an experimental study, Porras et al. [102] transferred genetically modified material from a high-fat diet (HFD)-fed donor mouse and a QE-fed mouse into HFD-fed recipient mice. This resulted in a protective phenotype against metabolic syndrome-associated liver disease (MASLD) compared to the HFD-fed donor, and this improvement was associated with alterations in the gut-hepatic axis in the recipients. Petrov et al. [106] demonstrated that the protective gut microflora (dHFDQ) influenced the bile acid profile in feces and plasma by promoting the synthesis of several secondary bile acids. This effect was associated with the prevention of MASLD in HFD-fed recipients. These studies have demonstrated the complex interplay between QE, the gut-liver axis and bile acids in the protection against MASLD.

### 3.6. Effects on the Senescence of Cells

During senescence, cellular changes such as telomere shortening, nuclear enlargement and genomic and mitochondrial DNA damage occur. Cellular senescence is defined as a state of permanent cell cycle arrest in which cells no longer undergo mitosis but remain metabolically active, often secreting pro-inflammatory cytokines, chemokines, and proteases. This phenomenon, commonly referred to as the senescence-associated secretory phenotype (SASP), plays a significant role in the process of ageing and the development of various diseases, including MASLD. The accumulation of senescent cells in tissues contributes to the development of an inflammatory response and the subsequent dysfunction of the tissue, which in turn serves to exacerbate the progression of the disease. In the context of MASLD, the accumulation of senescent hepatocytes has been demonstrated to exacerbate liver damage and metabolic dysfunction, thereby contributing to the progression of liver steatosis. A study demonstrated that the induction of senescence in hepatocytes via a DNA repair defect resulted in the development of liver steatosis, thereby further establishing the role of senescence in the progression of MASLD [107]. The results of research studies indicate that therapies that target senescent cells have the potential to alleviate disease symptoms, improve outcomes and reduce inflammation while restoring tissue function. In this context, the combination of dasatinib and QE has been identified as a highly efficacious anti-senescence therapy. Dasatinib, a tyrosine kinase inhibitor, and QE, have been demonstrated to act in a synergistic manner, eliminating senescent cells and attenuating the pro-inflammatory environment created by the SASP [108].

The administration of QE has been demonstrated to downregulate the cellular senescence pathway. This finding suggests that QE exerts its anti-senescence effects by modulating the underlying pathways that promote cell cycle arrest and SASP. The precise molecular mechanisms through which QE modulates these pathways remain a topic of active research. However, it is hypothesized that QE may affect pathways such as protein p53/p21 and protein p16INK4a, which are known to regulate cellular senescence. In addition to its gene-modulating effects, QE has been demonstrated to enhance metabolic outcomes in models of metabolic disease. Ogrodnik et al. [109] reported that treatment with dasatinib and QE significantly reduced the expression of senescence-associated markers and suppressed liver fat deposition in a model of MASLD with a mutated leptin receptor. These findings highlight the potential of QE to not only reduce the burden of senescent cells but also to improve the metabolic dysfunction associated with liver diseases such as MASLD.

### 3.7. Compensation of MASLD by Other Mechanisms

An increasing body of evidence from scientific studies suggests that bile acids play a significant role in the development of MASLD. Studies demonstrated the pivotal role of FXR and TGR5 in regulating bile acid function within the liver [110]. In mice with a mutated leptin receptor, treatment with QE led to an increase in the expression levels of FXR and TGR5, which subsequently resulted in the normalization of abnormal serum and liver bile acid levels. In vitro studies demonstrated that QE exerts analogous effects in HepG2 cells exposed to free fatty acids. This finding is evidenced by the improvement of the FXR/TGR5 axis activation and the normalization of fatty deposits [111]. Moreover, the authors demonstrated that the administration of QE resulted in reduced hepatic edema, the normalization of liver enzymes, reduced hyperglycemia and lipid accumulation in the liver of leptin receptor mutated mice. The authors emphasized the potential of QE in mitigating type 2 diabetes-induced MASLD through its antioxidant and anti-inflammatory properties, as well as its ability to enhance lipid metabolism [111]. Zhang et al. [112] showed QE enhances the activity of hepatic cholesterol 7α-hydroxylase, a pivotal enzyme involved in the conversion of cholesterol to bile acids that in rats fed an AIN-93 diet. The aforementioned studies indicate that QE reduced hepatic steatosis in MASLD by regulating bile acid metabolism and activating the FXR/TGR5 axis in the liver.

The dysregulation of DNL has recently been the subject of considerable research interest as a critical factor in the development of MASLD. In a study conducted by Gnoni et al. [113] it was observed that the simultaneous presence of elevated glucose and free fatty acid levels in a cellular model of MASLD resulted in the highest accumulation of cellular lipids. An increased expression of the mitochondrial citrate carrier (CiC), cytosolic acetyl-CoA carboxylase 1 (ACACA), and diacylglycerol acyltransferase 2 (DGAT2) was revealed, these are involved in fatty acid and triglyceride synthesis, respectively. XBP-1, a marker of endoplasmic reticulum stress, along with SREBP-1, were identified as transcription factors associated with DNL activation. The administration of QE was observed to reduce lipid accumulation and the expression of SREBP-1 and XBP-1, in addition to their lipogenic target genes, in cells exhibiting steatosis. The anti-lipogenic effect of QE is primarily mediated by the strong phosphorylation of ACACA, which serves as the initial catalytic step in the DNL pathway. The elevated level of ACACA phosphorylation in QE-treated cells was attributed to the involvement of AMPK, along with a reduction in PP2A phosphatase activity. Collectively, these findings underscore the direct anti-lipogenic impact of QE, which is mediated by its action on the ACACA/AMPK/PP2A axis, thereby indicating that QE represents a promising candidate for the management of MASLD.

It is also possible that other mechanisms involved in the MASLD progression may be regulated by QE, although this possibility has yet to be fully explored. For instance, there was a positive correlation between iron overload and the severity of MASLD. The iron-chelating properties of QE are notable, due to the presence of numerous free hydroxyl groups [114]. It was demonstrated that QE can prevent ethanol-induced liver iron overload in males, including excessive hepatic labile iron pool and lysosomal iron retention [115]. Nevertheless, it remains unclear whether this mechanism is involved in the pathogenesis of MASLD. In vivo and in vitro studies are listed in Table 1.

## 4. Effect of Quercetin on MASLD—Clinical Trials

The number of clinical studies on the therapeutic effects of QE on MASLD is limited. The impact of QE on blood biochemical markers and pro- and anti-inflammatory cytokines in patients with MASLD was examined by Prysyazhnyuk and Voloshyn [129]. This study examined the effect of a 2-week regimen of QE in conjunction with basal therapy on serum levels of AST, ALT and glutamate aminotransferase (GGT). The levels of the enzymes alanine ALT and GGT were significantly reduced by 37.2%, 50.4% and 89.9%, respectively. Furthermore, the levels of total cholesterol, triglycerides, and TNF-α were significantly decreased by 16.7%, 33.3% and 39.8%, respectively. These findings indicate that QE may possess the potential for therapeutic value in the treatment of MASLD.

The primary objective of the recently initiated randomized double-blind clinical trial (ID: NL-OMON53809) [130] is to determine the effect of dasatinib and QE on liver fibrosis, as assessed histologically, in patients with MASLD and biopsy-proven fibrosis. The primary endpoint will be a binary score of at least a one point improvement in fibrosis and an MASLD score on histology at 21 weeks. The secondary objectives are to investigate the effects of dasatinib and QE on a number of additional parameters, including liver-related histological parameters, biomarkers associated with MASLD, glucose metabolism, pulse, blood pressure, electrocardiogram, hematological and biochemical parameters as well as the composition of the gut microbiota (at baseline and at the conclusion of the treatment period, which will be at week 21). Furthermore, the safety of dasatinib and QE will be evaluated. In this clinical trial, 30 patients with MASLD will be treated with dasatinib plus QE (1000 mg QE +100 mg dasatinib or placebo) intermittently three days per week for three weeks, followed by a four-week drug-free period. This treatment cycle will be repeated three times, thus for a total of 21 weeks.

Although there is no established standard dosage for QE in the context of MASLD, clinical studies have typically employed doses ranging from 500 mg to 1000 mg per day. Clinical trials [129] have reported the potential occurrence of side effects, including gastrointestinal discomfort, headaches, and kidney toxicity, in individuals administered quercetin. Therefore, the dosage of quercetin appropriate for MASLD patients should be carefully considered, and healthcare providers should monitor for adverse effects, particularly in individuals with pre-existing conditions or those taking other medications. It is important to note, however, that quercetin’s bioavailability can be low when taken orally. Enhancing its absorption may therefore be beneficial, for example by combining it with other compounds like bromelain or taking it in liposomal form. This action may improve its therapeutic potential. Given its antioxidant, anti-inflammatory, lipid-regulating, and anti-fibrotic properties, QE represents a promising natural compound for the management of MASLD. While lifestyle modifications remain the cornerstone of MASLD treatment, QE may offer supplementary benefits by targeting the underlying metabolic and molecular processes that drive the disease. As research progresses, quercetin could emerge as a crucial complementary therapy in the fight against MASLD, helping to alleviate the growing burden of this condition on global health.

## 5. Conclusions and Perspectives

MASLD is one of the most prevalent chronic liver conditions globally. It is characterized by the excessive accumulation of fat in the liver, despite the absence of significant alcohol consumption. There is a growing emphasis on the development of efficacious treatments and preventive strategies for MASLD. Quercetin, a naturally occurring flavonoid found in a wide variety of fruits and vegetables, has recently garnered attention for its potential therapeutic benefits in NAFLD, largely due to its antioxidant, anti-inflammatory, and lipid-modulating properties. A number of preclinical studies utilizing animal models of MASLD have demonstrated the efficacy of quercetin in improving liver health. For instance, the administration of quercetin to models of MASLD induced by a high-fat diet has been observed to result in a reduction in liver fat content, inflammation, and oxidative stress markers. Furthermore, quercetin was observed to improve liver enzyme levels, including alanine aminotransferase and aspartate aminotransferase, which are frequently elevated in liver disease.

Nevertheless, the number of clinical studies examining the effects of quercetin in human patients with MASLD is relatively limited. The results of preliminary trials indicate that quercetin supplementation may be an effective approach for improving liver enzyme levels, insulin resistance, and overall metabolic health in patients with MASLD. Nevertheless, the trials are frequently limited by small sample sizes, short durations, and variable dosages, which collectively impede the ability to draw definitive conclusions. Further well-designed, large-scale clinical trials are required to establish the efficacy and safety profile of quercetin in the long-term management of MASLD. Quercetin is generally considered safe when consumed through dietary sources; however, the safety of high-dose quercetin supplements, particularly for long-term use, remains a concern. Future research should therefore focus on the following: conducting large-scale, randomized controlled trials to assess quercetin’s long-term efficacy and safety in MASLD patients; investigating the optimal dosage and formulation of quercetin for therapeutic use; and exploring quercetin’s potential synergistic effects when combined with other natural compounds or existing MASLD treatments.

## Figures and Tables

**Figure 1 molecules-29-05245-f001:**
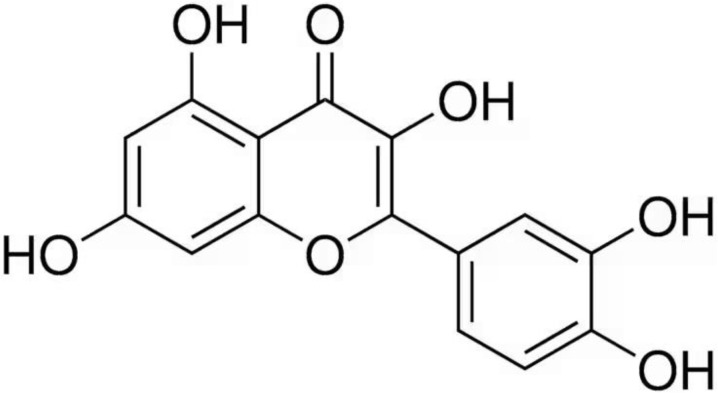
Chemical formula of quercetin.

**Figure 2 molecules-29-05245-f002:**
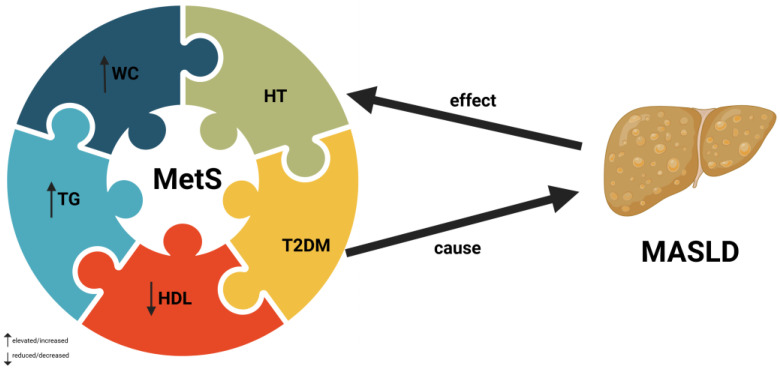
Bidirectional relationship between MetS and MASLD. Abbreviations: HDL—high-density lipoproteins, HT—hypertension, MASLD—metabolic dysfunction-associated steatotic liver disease, MetS—metabolic syndrome, TG—triglycerides, T2DM—type 2 diabetes mellitus, WC—waist circumference.

**Figure 3 molecules-29-05245-f003:**
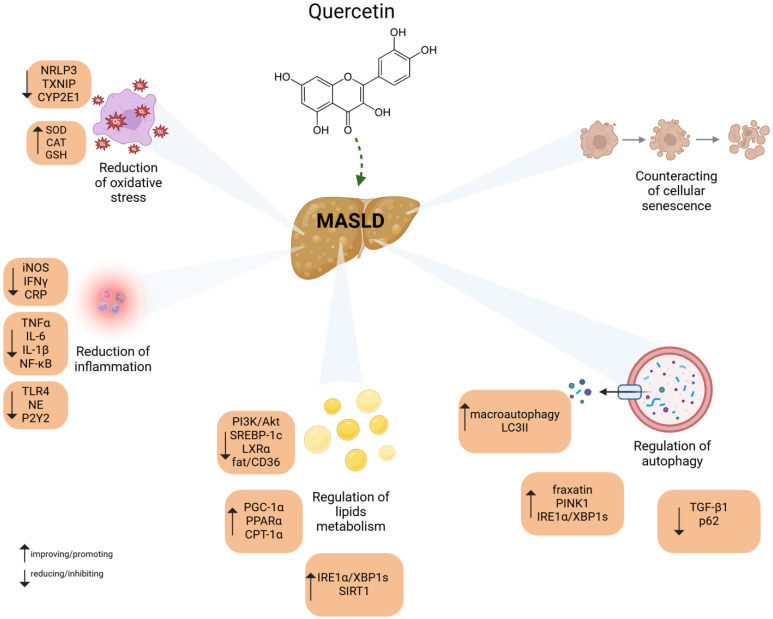
Molecular mechanisms of quercetin action in MASLD. For further explanations please see text below. All of the abbreviations are listed in the end of manuscript.

**Figure 4 molecules-29-05245-f004:**
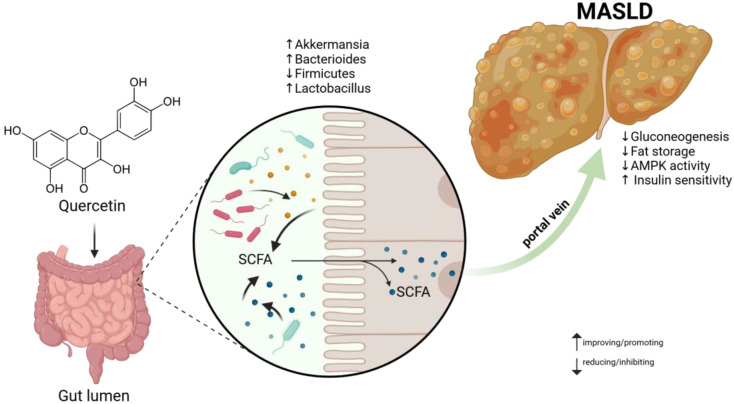
Scheme of the mechanism of action of dietary polyphenols in the treatment of MASLD using quercetin as the example. Polyphenols attenuate MASLD by many pathways to inhibit mitochondrial malfunction and/or reduce the lipid accumulation. By maintaining proper Firmicutes to Bacterioidetes ratio, the correct ratio of SCFAs is preserved. SCFAs transported to the liver by the portal vein in appropriate amounts have beneficial effects on MASLD. Adopted and modified from the figures by Yang et al. 2022 [99]. Abbreviations: AMPK—AMP-activated protein kinase, MASLD—metabolic dysfunction-associated steatotic liver disease, SCFA—short-chain fatty acid.

**Table 1 molecules-29-05245-t001:** Hepatoprotective effects of QE against MASLD—in vivo and in vitro studies.

Model	Dose	Effects	References
In vivo studies			
Rat	20 mg/kg	↓ALT, AST	[81]
Rat	50 and 100 mg/kg	↓ALT, AST ↓Serum and hepatic TC and TG ↓Lipid vacuoles ↓Inflammation	[116]
Rat	50 mg/kg	↓AST, AST ↓Serum TC and TG ↓Lipid accumulation ↓Inflammation	[85]
Mice	100 mg/kg	↓Hepatic and serum TC, TG ↓Lipid accumulation	[96]
Mice	50 mg/kg	↓ALT ↓Serum TG ↓Hepatic TG and FFAs ↓Lipid accumulation ↓Inflammation ↓IL-6	[101]
Mice	80 mg/kg	↑BAs ↓Lipogenic pathways	[106]
Mice	50 mg/kg (+5 mg/kg of Dasatinib)	↑Glucose tolerance ↑Insulin sensitivity ↓Inflammation	[108,109]
Mice	20 mg/kg	→Lipid vacuoles and lipid accumulation	[117]
Mice	25 mg/kg	↓Serum and hepatic TG ↓Lipid accumulation	[90]
Mice	330 mg/kg	↓Lipid accumulation	[118]
Mice	330 mg/kg	↓Serum TG ↓Lipid accumulation	[87]
Mice	50 mg/kg	↓Serum and hepatic TC and TG ↓Lipid vacuoles	[89]
Mice	50 mg/kg	↓ALT ↓Plasma and hepatic TC and TG ↓Plasma FFAs ↓Plasma TNFα, MCP-1	[119]
Mice	100, 500 mg/kg	↓Apoptosis	[120]
Mice	50 mg/kg	↓AST, ALT ↓Serum TG ↓Lipid vacuoles ↓Inflammation	[121]
Mice		→Plasma cytokines ↓Inflammatory cytokines	[122]
Mice	50 mg/kg	↓ALT, AST ↓Lipid vacuoles ↓Inflammation	[123]
Rat	75 mg/kg	↓Lipid accumulation ↓Inflammation	[124]
Rat	800 mg/kg	→ALT, AST	[125]
Rat	50 mg/kg	↓Serum and hepatic TG ↓Lipid vacuoles	[126]
Rat	10 and 50 mg/kg	↓Serum TC and TG	[93]
In vitro studies			
HepG2	100 µM	↓TG ↓IL-8, TNF-α	[31]
Primary hepatocytes	50 and 100 μM	↓TG	[94]
HepG2	10 µM	↓TG	[95]
HepG2	100 μM	↓TG ↓Lipid accumulation	[96]
Huh7.5	50 µM	↓TG ↓Lipid accumulation	[127]
HepG2	1, 10 and 100 µM	↓Lipid accumulation	[128]
Primary hepatocytes, RHPCs, HLO2, HepG2	20 µM	↓NLRP3, IL-1β, IL-18	[79]
Primary rat hepatocytes	40 µM	↓TG ↓Lipid accumulation	[80]

↑ upregulated, ↓ downregulated, → unchanged.

## Data Availability

Data are available upon a reasonable request from the corresponding author.

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
