# Peer review of "Quercetin: A Promising Candidate for the Management of Metabolic Dysfunction-Associated Steatotic Liver Disease (MASLD)"

_molecules, 2024, doi:10.3390/molecules29225245_

Round 1
Reviewer 1 Report
Comments and Suggestions for Authors
The authors well summarized the available information on the therapeutic effects of QE in MASLD.
In section 3.3., the action of QE on PPARgamma in the liver is described. As is well known, there are two lipid synthesis systems in the liver, one by SREBP-1c and the other via PPARgamma. In humans, PPARgamma has been reported to play an important role in fatty liver development. In 2.1. write about PPARgaama-mediated fatty liver development.
Author Response
The authors would like to thank the Reviewer for valuable comments which have helped to improve the quality of the manuscript. We hope that the revisions in the manuscript will be sufficient to make our manuscript suitable for publication. PPARγ-mediated fatty liver development is described in section 2.1.
Reviewer 2 Report
Comments and Suggestions for Authors
The article presented for review concerns the properties of quercetin and its role in metabolic dysfuncion-associated steatotic liver disease (MASLD). The article is well written and well organized. It is a valuable source of information on the therapeutic effects of quercetin in MASLD. In my opinion, the article can be published in a journal. Minor comments should be taken into account:
- the abbreviations used are not always explained at the place of their first use. The authors should carefully review the manuscript and check this.
- Since the article contains a lot of abbreviations, it seems that preparing a list of the abbreviations used and placing them at the beginning or end of the article would make it easier to read
- in the introduction, when describing quercetin, its structural formula should be included
Author Response
The authors would like to thank the Reviewer for valuable comments which have helped to improve the quality of the manuscript. We hope that the revisions in the manuscript and our accompanying responses will be sufficient to make our manuscript suitable for publication.
The article presented for review concerns the properties of quercetin and its role in metabolic dysfuncion-associated steatotic liver disease (MASLD). The article is well written and well organized. It is a valuable source of information on the therapeutic effects of quercetin in MASLD. In my opinion, the article can be published in a journal. Minor comments should be taken into account:
- the abbreviations used are not always explained at the place of their first use. The authors should carefully review the manuscript and check this.
Thank you for comment. The manuscript has been double-checked independently by two of co-authors and explanations of used abbreviations have been included where necessary.
- Since the article contains a lot of abbreviations, it seems that preparing a list of the abbreviations used and placing them at the beginning or end of the article would make it easier to read
Thank you for comment. A list of all of the used abbreviations and their explanations has been placed in the end of the manuscript.
- in the introduction, when describing quercetin, its structural formula should be included
Thank you for comment. The structural formula of quercetin has been implemented into Introduction section.
Reviewer 3 Report
Comments and Suggestions for Authors
In this manuscript, the authors have reviewed the protective effect and mechanism of quercetin against metabolic dysfunction-associated steatotic liver disease (MASLD). Overall, this review is of good interest, while it is not well organized. Some issues should be clarified:
(1) In line 79-84, the authors should add the reference for this paragraph.
(2) In line 85-113, why the authors mentioned the metabolic syndrome (MetS) here? This review should focus on MASLD rather than MetS.
(3) In line 125-127, “The comprehensive discussion on molecular and ...... regarding causes of MASLD” need to be deleted.
(4) In the part of “2. Pathogenesis of MASLD”, the authors need to draw several graphs to summarize lipids metabolism and insulin signaling; oxidative stress, inflammation and lipotoxicity; autophagy; and gut microbiota. Why the authors didn’t mention mitochondria in this part? In addition, the sections of “2.3 autophagy” and “2.4 gut microbiota” were not written in depth.
(5) In the section of “3.1. Precluding oxidative stress”, the first paragraph is about mechanism, it should be written in the part of “2. Pathogenesis of MASLD”; the second part is about polyphenols, and need to be deleted here.
(6) In line 301-303, “Furthermore, QE effectively protected HepG2 cells from ...... cells against oleic acid-induced ROS increase” is about ROS, and need to deleted in the section of “3.2. Influence on inflammatory processes”.
(7) In the section of “3.3. Regulation of lipid metabolism and mitochodrial dysfunction”, the logical structure should be reorganized. In line 337, why the authors mentioned autophagy here?
(8) In the section of “3.4. Regulation of autophagy”, why the authors mentioned “ Hepatocellular carcinoma (HCC)” here?
(9) In the section of “3.5. Modulation of gut microbiota composition”, why the authors didn’t mention bile acids here?
(10) In line 379-380, the authors mentioned “antioxidant levels, inflammatory pathways, and β-oxidation”, while these were not presented in Figure 2.
(11) The part of “3. Mechanism of action of quercetin in MASLD” and “4. Quercetin and MASLD - in vitro and in vivo studies” should combined into one part. In addition, the authors need to draw several graphs to summarize the protective mechanisms of quercetin here.
(12) In line 724-727, why the authors only listed the studies not described in the text in Table 1? All the in vitro and in vivo studies should be listed in table 1, and the mechanisms should be added here.
Author Response
The authors would like to thank the Reviewer for valuable comments which have helped to improve the quality of the manuscript. We hope that the revisions in the manuscript and our accompanying responses will be sufficient to make our manuscript suitable for publication.
In this manuscript, the authors have reviewed the protective effect and mechanism of quercetin against metabolic dysfunction-associated steatotic liver disease (MASLD). Overall, this review is of good interest, while it is not well organized. Some issues should be clarified:
(1) In line 79-84, the authors should add the reference for this paragraph.
Thank you for your comment. The reference has been included.
(2) In line 85-113, why the authors mentioned the metabolic syndrome (MetS) here? This review should focus on MASLD rather than MetS.
Thank you for suggestion. Information on MetS has been removed from the text.
(3) In line 125-127, “The comprehensive discussion on molecular and ...... regarding causes of MASLD” need to be deleted.
Thank you for comment. The sentence has been deleted.
(4) In the part of “2. Pathogenesis of MASLD”, the authors need to draw several graphs to summarize lipids metabolism and insulin signaling; oxidative stress, inflammation and lipotoxicity; autophagy; and gut microbiota. Why the authors didn’t mention mitochondria in this part? In addition, the sections of “2.3 autophagy” and “2.4 gut microbiota” were not written in depth.
Thank you for comment. A Figure 3 that briefly representing reviewed causes of MASLD and potential of quercetin has been implemented according to Reviewer’s suggestion. The sections 2.3 and 2.4 have been improved and expanded as well as 3.5, where also important information regarding gut microbiota has been included, yet the role of mitochondria in the pathogenesis of MASLD has been implemented in section 2.3.
(5) In the section of “3.1. Precluding oxidative stress”, the first paragraph is about mechanism, it should be written in the part of “2. Pathogenesis of MASLD”; the second part is about polyphenols, and need to be deleted here.
The section entitled „Precluding oxidative stress” has been modified according to the Reviewer's recommendations.
(6) In line 301-303, “Furthermore, QE effectively protected HepG2 cells from ...... cells against oleic acid-induced ROS increase” is about ROS, and need to deleted in the section of “3.2. Influence on inflammatory processes”.
Thank you for suggestion. The sentence has been deleted.
(7) In the section of “3.3. Regulation of lipid metabolism and mitochodrial dysfunction”, the logical structure should be reorganized. In line 337, why the authors mentioned autophagy here?
The section entitled „Regulation of lipid metabolism and mitochondrial dysfunction” has been modified and reorganized to the reviewer's recommendations. Information on autophagy has been removed from this section.
(8) In the section of “3.4. Regulation of autophagy”, why the authors mentioned “ Hepatocellular carcinoma (HCC)” here?
Thank you for comment. The information on HCC has been deleted from this section.
(9) In the section of “3.5. Modulation of gut microbiota composition”, why the authors didn’t mention bile acids here?
Thank you for suggestion. Information conerning bile acid have been added to section “3.5. Modulation of gut microbiota composition”.
(10) In line 379-380, the authors mentioned “antioxidant levels, inflammatory pathways, and β-oxidation”, while these were not presented in Figure 2.
Thank you for your comment. Indeed, the caption under Fig. 2 contained redundant information, which has now been removed.
(11) The part of “3. Mechanism of action of quercetin in MASLD” and “4. Quercetin and MASLD - in vitro and in vivo studies” should combined into one part. In addition, the authors need to draw several graphs to summarize the protective mechanisms of quercetin here.
Thank you for suggestion. The sections entitled “3. Mechanism of action of quercetin in MASLD” and “4. Quercetin and MASLD - in vitro and in vivo studies” have been combined into one part according to the Reviewer's recommendations. Furthermore, Figure 3 that briefly summarize reviewed quercetin influence on MASLD has been implemented.
(12) In line 724-727, why the authors only listed the studies not described in the text in Table 1? All the in vitro and in vivo studies should be listed in table 1, and the mechanisms should be added here.
Thank you for suggestion. Now, all the in vitro and in vivo studies have been listed in Table 1.
Reviewer 4 Report
Comments and Suggestions for Authors
In this review, the authors describe the Management of Metabolic Dysfunction-Associated Steatotic Liver Disease (MASLD) and place quercetin as one of the alternatives for controlling this disease. Mention quercetin with treatment with this flavonoid from a natural source and has been demonstrated to possess a number of beneficial physiological effects, including anti-inflammatory, anti-cancer and antifungal properties. Additionally, it functions as a natural antioxidant. Preclinical evidence indicates that quercetin may play a beneficial role in reducing liver damage and improving metabolic health. However, the content is interesting and well documented and discussed.
The results are well described and referenced with current publications.The figures and tables are well discussed and quality of Figures is good.
In conclusion: The authors concluded that the number of clinical studies examining the effects of quercetin in human patients with MASLD is relatively limited. The results of preliminary trials indicate that quercetin supplementation may be an effective approach for improving liver enzyme levels, insulin resistance, and overall metabolic health in patients with MASLD.
Author Response
The authors would like to thank you for your kind assessment of our work. We appreciate the time and effort you have put into your review.
Round 2
Reviewer 3 Report
Comments and Suggestions for Authors
The authors have solved my concerns well, and this manuscript can be accepted now.